# Proposal of Quick Diagnostic Criteria for Disseminated Intravascular Coagulation

**DOI:** 10.3390/jcm11041028

**Published:** 2022-02-16

**Authors:** Hideo Wada, Akitaka Yamamoto, Masaki Tomida, Yuhuko Ichikawa, Minoru Ezaki, Jun Masuda, Masamichi Yoshida, Shunsuke Fukui, Isao Moritani, Hidekazu Inoue, Katsuya Shiraki, Kei Suzuki, Hiroshi Imai, Motomu Shimaoka, Hideto Shimpo

**Affiliations:** 1Department of General and Laboratory Medicine, Mie Prefectural General Medical Center, Yokkaichi 510-0885, Japan; 2Department of Emergency and Critical Care Center, Mie Prefectural General Medical Center, Yokkaichi 510-0885, Japan; akitaka-yamamoto@mie-gmc.jp (A.Y.); st25053@yahoo.co.jp (M.T.); 3Department of Central Laboratory, Mie Prefectural General Medical Center, Yokkaichi 510-0885, Japan; ichi911239@yahoo.co.jp (Y.I.); ajbyd06188@yahoo.co.jp (M.E.); 4Department of Cardiovascular Medicine, Mie Prefectural General Medical Center, Yokkaichi 510-0885, Japan; jun-masuda@mie-gmc.jp; 5Department of Respiratory Medicine, Mie Prefectural General Medical Center, Yokkaichi 510-0885, Japan; masamichi-yoshida@mie-gmc.jp; 6Department of Gastroenterology, Mie Prefectural General Medical Center, Yokkaichi 510-0885, Japan; m13092sf@jichi.ac.jp (S.F.); isao-moritani@mie-gmc.jp (I.M.); hidekazu-inoue@mie-gmc.jp (H.I.); katsuya-shiraki@mie-gmc.jp (K.S.); 7Emergency Critical Care Center, Mie University Graduate School of Medicine, Tsu 514-8507, Japan; keis@med.mie-u.ac.jp (K.S.); hi119@clin.medic.mie-u.ac.jp (H.I.); 8Department of Molecular Pathobiology and Cell Adhesion Biology, Mie University Graduate School of Medicine, Tsu 514-8507, Japan; motomushimaoka@gmail.com; 9Mie Prefectural General Medical Center, Yokkaichi 510-0885, Japan; hideto-shimpo@mie-gmc.jp

**Keywords:** DIC, diagnosis, PT-INR, platelet count, D-dimer

## Abstract

Background. The diagnostic criteria for disseminated intravascular coagulation (DIC) vary and are complicated and the cut-off values are different. Simple and quick diagnostic criteria for DIC are required in physicians for critical care. Material and methods. Platelet counts, prothrombin time–international normalized ratio (PT-INR) and D-dimer levels were examined in 1293 critical ill patients. Adequate cut-off values of these parameters were determined and a quick DIC score using these biomarkers was proposed. The quick DIC score was evaluated using a receiver operating characteristic (ROC) analysis. Results. Using the Japanese Ministry of Health, Labor and Welfare diagnostic criteria, 70 and 109 patients were diagnosed with DIC and pre-DIC, respectively. The ROC analysis of factors difference between DIC and non-DIC, revealed the following cut-off values: PT-INR, 1.20; platelet count, 12.0 × 10^10^/L and D-dimer, 10.0 μg/mL. Based on the above results, the quick DIC score system was proposed. All patients with DIC had a quick DIC score of 3, 4 or 5, and 85.3% of the patients with pre-DIC had a quick DIC score of ≥3 points. All patients with pre-DIC had a score of ≥2 points. In the ROC analysis, the area under the curve was 0.997 for DIC vs. non-DIC, and 0.984 for pre-DIC + DIC vs. non-DIC, and the cut-off value was 3 points for DIC and 2 points for DIC + pre-DIC. The quick DIC scores of non-survivors were significantly higher than those of survivors. Conclusions. The Quick DIC score system is a simple and useful tool that can be used for the diagnosis of DIC and pre-DIC. Further evaluation of the quick DIC score system in a large-scale study is required.

## 1. Introduction

Disseminated intravascular coagulation (DIC) is frequently associated with infectious diseases, hematological malignancy, and solid cancer, and causes organ failure and bleeding symptoms, often resulting in poor outcomes [1,2]. The DIC patients are generally treated for underlying diseases, and those with major bleeding are treated with supplementary therapy from blood products [3]. In Japan, where an early treatment of DIC has been recommended [4], DIC patients have been treated with antithrombin [5] or recombinant thrombomodulin [6]. Therefore, diagnostic criteria for DIC have been established by the Japanese Ministry of Health, Labor and Welfare (JMHLW) [7], the International Society of Thrombosis Haemostasis (ISTH) [8], the Japanese Association for Acute Medicine (JAAM) [9], and the Japanese Society of Thrombosis Hemostasis (JSTH) [10].

These diagnostic criteria for DIC are based on scoring systems that include the platelet counts, prothrombin time (PT), fibrin-related markers, and fibrinogen [11]. The scoring system for DIC may be complicated in practice and fibrin-related markers require standardization [12]. Multiple cut-off values of the parameters are used among the four diagnostic criteria for DIC [11]. In the emergency room, more simple diagnostic criteria are required to facilitate the early treatment of DIC. The recently developed sepsis-induced coagulopathy (SIC) score, which includes PT, platelet count, and the sequential organ failure assessment score, may be a simple and easy score, and may be useful for diagnosing DIC in sepsis patients [13]. The cut-off values of fibrin-related markers have not been established, varying among reports for DIC using ISTH overt-DIC diagnostic criteria [12]. Furthermore, the JAAM diagnostic criteria for DIC are not useful for diagnosing DIC due to non-infectious diseases, and the JSTH diagnostic criteria for DIC are complicated. Therefore, we diagnosed DIC using the JMHLW diagnostic criteria for DIC.

In this study, we determined adequate cut-off values of PT, D-dimer, and platelet count for the diagnosis of DIC and propose a quick DIC score system.

## 2. Materials and Methods

The study population included patients with the following conditions who were managed at Mie Prefectural General Medical Center from 1 September 2019 to 28 December 2020: sepsis (*n* = 58), critical illness without sepsis (*n* = 152), pneumonia (*n* = 107), obstetric disease (*n* = 16), solid cancer (*n* = 42), aneurysm (*n* = 78), other respiratory disease (*n* = 28), peripheral arterial and venous thrombosis (PAVTE, *n* = 44), hematological disorder (*n* = 41), other infection (*n* = 77), other digestive disorder (*n* = 87), cerebrovascular disorder (*n* = 176), acute coronary syndrome (*n*= 67), other heart disease (*n* = 94), urinary tract disease (*n* = 125), unidentified clinical syndrome (*n* = 125), and other conditions (*n* = 50). DIC was diagnosed using the Japanese Ministry of Health Labor and Welfare criteria for DIC (Appendix A) [7]. Patients with a DIC scores of ≥7 points, with 5–6 points, and ≤4 points were diagnosed with DIC, pre-DIC, and non-DIC, respectively. Cerebrovascular disorder was diagnosed by computed tomography or magnetic resonance imaging; acute coronary syndrome was diagnosed by coronary angiography, electrocardiography, and the detection of elevated troponin; and PAVTE was diagnosed using computed tomography or venous ultrasound. The study protocol (2019-K9) was approved by the Human Ethics Review Committee of Mie Prefectural General Medical Center, and informed consent was obtained from each participant. This study was carried out in accordance with the principles of the Declaration of Helsinki.

D-dimer was measured using LPIA-Genesis (LSI Medience, Tokyo, Japan), with a STACIA system (LSI Medience). The prothrombin time (PT) international normalized ratio (INR) was measured by a Thromborel S (Sysmex Co., Kobe, Japan) using an automatic coagulation analyzer CS-5100 (Sysmex Co.). The platelet counts were measured using a full-automatic blood cell counter XN-3000 (Sysmex Co.). The median (2.5–97.5 percentile) values of D-dimer, PT-INR, and platelet counts were 0.4 μg/mL; 0.1–0.5 μg/mL, 0.96; 0.90–1.04, and 21.9 × 10^10^/L; 17.9–25.9 × 10^10^/L, respectively.

### Statistical Analyses

The data are expressed as the median (25–75th percentile). The significance of differences between groups was examined using the Mann–Whitney *U-*test. A multiple regression analysis of PT-INR, D-dimer, and platelet counts for diagnosing DIC or pre-DIC vs. non-DIC was conducted. The cut-off values were determined as the point at which the sensitivity curve and specificity curve intersected, and were examined by a receiver operating characteristic (ROC) analysis. *p* values of <0.05 were considered to indicate a statistically significant difference. All of the statistical analyses were performed using the Stat-Flex software program (version 6; Artec Co., Ltd., Osaka, Japan).

## 3. Results

Regarding the evaluation of patients using the JMHLW diagnostic criteria, 70 and 109 patients were diagnosed with DIC and pre-DIC, respectively. Sepsis, critical illness without sepsis, pneumonia, obstetric disease, and solid cancer were frequently associated with DIC (Table 1).

The PT-INR in patients with DIC (1.61: 1.34–1.96) was significantly higher in comparison to those with pre-DIC (1.24: 1.06–1.38) or non-DIC (1.01: 0.94–1.10), and in patients with pre-DIC than in those with non-DIC (Figure 1a). The results of a multiple regression analysis of PT-INR, D-dimer, and platelet counts for diagnosing DIC or pre-DIC vs. non-DIC are shown in Table 2. Standard β value was higher in the order of D-dimer, platelet count, and PT-INR. Regarding the ROC analysis of DIC vs. non-DIC, the adequate cut-off value was 1.20, the sensitivity and specificity were 88.8%, and the area under the curve (AUC) and negative predictive value (NPV) were 0.934 and 98.9% (Table 3).

The platelet count was significantly lower in patients with DIC (8.3 × 10^10^/L: 5.2–11.4 × 10^10^/L) than in those with pre-DIC (12.6 × 10^10^/L: 9.2–20.3 × 10^10^/L) or non-DIC (21.8 × 10^10^/L: 16.9–27.7 × 10^10^/L), and in those with pre-DIC than in those with non-DIC (Figure 1b). Regarding ROC analysis for DIC vs. non-DIC, the adequate cut-off value was 12.0 × 10^10^/L, and the sensitivity and specificity were 93.6% and 80.0%, and the AUC and NPV were 0.925 and 44.1%.

The D-dimer levels in patients with DIC (25.0 μg/mL: 16.1–47.8 μg/mL) were significantly higher than those in patients with pre-DIC (17.0 μg/mL: 8.2–25.7 μg/mL) or non-DIC (1.7 μg/mL: 0.7–4.8 μg/mL), and in those with pre-DIC than in those with non-DIC (Figure 1c). Regarding the ROC analysis of DIC vs. non-DIC, the adequate cut-off value was 10.0 μg/mL, the sensitivity and specificity were 93.1% and 92.5%, respectively, and the AUC and NPV were 0.971 and 99.6%.

Based on the above results, we propose the quick DIC score system (Table 4). When these patients were evaluated using the quick DIC score system, all patients with DIC had a quick DIC score of ≥3 points, 85.3% of patients with pre-DIC had a score of ≥3 points, and all patients with pre-DIC had a score of ≥2 points (Figure 2). Regarding the ROC analysis (Figure 3a,b), the AUC was 0.997 for DIC vs. non-DIC, and 0.984 for pre-DIC + DIC vs. non-DIC, and the cut-off value was 3 points for DIC and 2 points for DIC + pre-DIC (Table 4). The quick DIC scores of non-survivors (2.0 points: 1.0–3.0 points) were significantly higher than those of survivors (1 point: 0–1.0 points). Regarding the ROC analysis of survival (Figure 3c), the AUC was 0.685 and the adequate cut-off value was 2 points (Table 5).

## 4. Discussion

About five percent of patients, especially those with critical illness (e.g., sepsis, pneumonia, obstetric disease, solid cancer, and aneurysms) developed DIC in this study. Therefore, we propose that severe infections, hematological malignancy, solid cancer, aneurysm, obstetric diseases, and critical illness (e.g., trauma, shock and inflammation, multiple organ failure, etc.) are included as underlying diseases due to DIC in the quick DIC scoring system (this is similar to the underlying diseases of DIC in the JMHLW DIC diagnostic criteria for DIC) [7]. These diseases are also generally recognized as underlying diseases of DIC [8,9].

In the ROC analysis, the PT-INR, platelet counts, and D-dimer levels showed high AUC values, suggesting that these parameters are useful for the diagnosis of DIC or pre-DIC. Although D-dimer has the highest AUC and Stdβ among the three parameters, it requires standardization [12] and the SIC scoring system does not include D-dimer [13]. Although the PT-INR has a high AUC and NPV, it reflects the liver function and anticoagulation therapy [14]. Although the platelet count is the most common parameter assessed DIC and has a high AUC, it was found to have a low NPV in this study, suggesting that thrombocytopenia is frequently observed in patients without DIC [15]. Finally, there are no perfect parameters for diagnosing DIC. Therefore, we propose our quick DIC scoring system, which uses a combination of three parameters (e.g., PT-INR, platelet count, and D-dimer) to diagnose DIC. In addition, we propose a single cut-off value for each parameter instead of multiple cut-off values (as used in the JMHLW [7], ISTH [8], JAAM [9], and JSTH [10] diagnostic criteria for DIC), which results in a simple and quick scoring system that can be used for the diagnosis of DIC in the critical care setting.

Regarding the evaluation of the diagnosis of DIC using the quick DIC score system, 100% of the patients with DIC and 85.3% of the patients with pre-DIC who were diagnosed using JMHLW diagnostic criteria in this study were diagnosed with possible DIC using the quick DIC score system, which showed a markedly high AUC for the diagnosis of DIC. Therefore, large-scale studies are required to confirm the usefulness of this quick DIC score system for the diagnosis of DIC and pre-DIC. Regarding the survival of patients assessed using the quick DIC score system, approximately 50% of the patients diagnosed with possible DIC died, suggesting that the quick DIC score may predict poor outcomes as DIC patients still show poor outcomes and DIC is one of the most important causes of death in critical ill patients [3,16].

Finally, the most important point concerning the quick DIC system is its use as a reference point of underlying diseases due to the onset of DIC. The second point is a single cut-off value of PT and D-dimer which has a high NPV for diagnosing DIC. The third point is a single cut-off value of platelet counts, which has a high sensitivity for diagnosing DIC.

## 5. Conclusions

The quick DIC score, which is determined based on underlying diseases, the PT-INR, platelet count, and D-dimer levels, was able to diagnose DIC and predict poor outcomes.

## Figures and Tables

**Figure 1 jcm-11-01028-f001:**
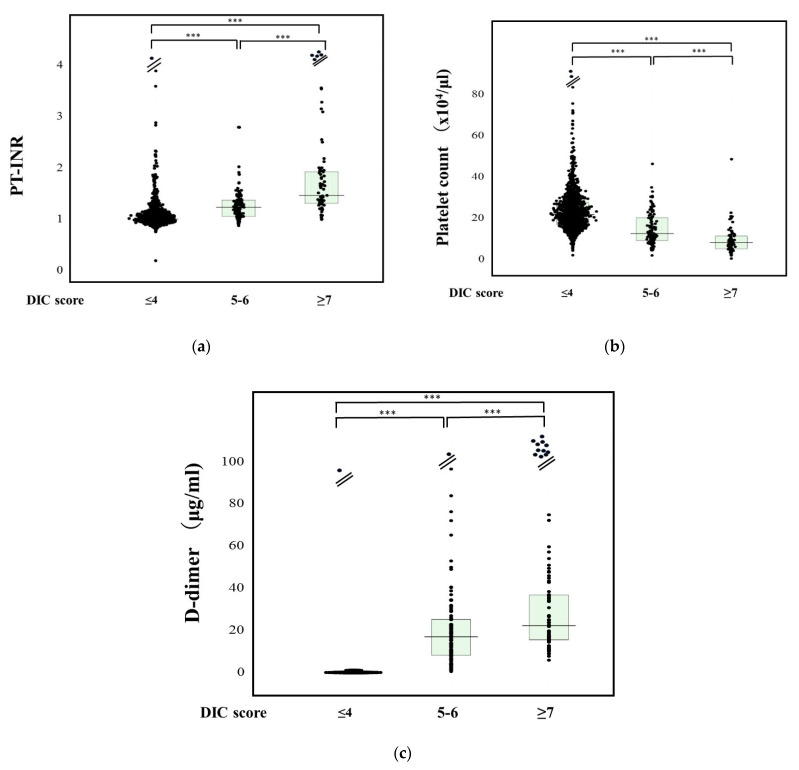
PT-INR (**a**), platelet counts (**b**), and D-dimer (**c**) DIC, disseminated intravascular coagulation; PT-INR, prothrombin time–international normalized ratio. ***, *p* < 0.001.

**Figure 2 jcm-11-01028-f002:**
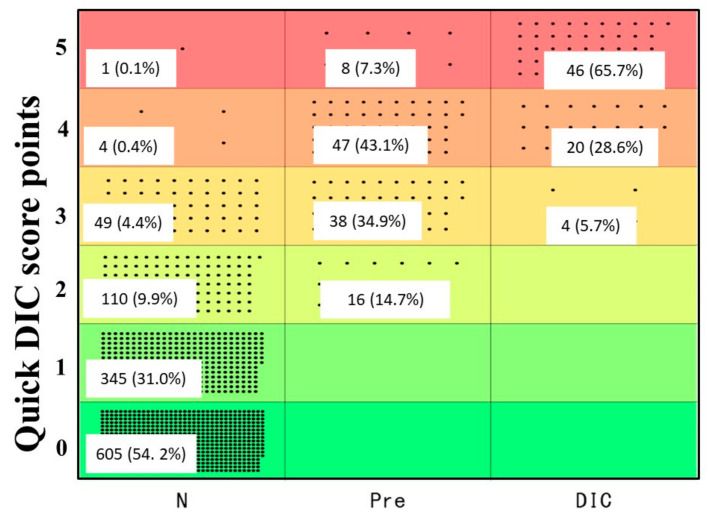
Quick DIC score in patients with non-DIC, pre-DIC, or DIC. DIC, disseminated intravascular coagulation.

**Figure 3 jcm-11-01028-f003:**
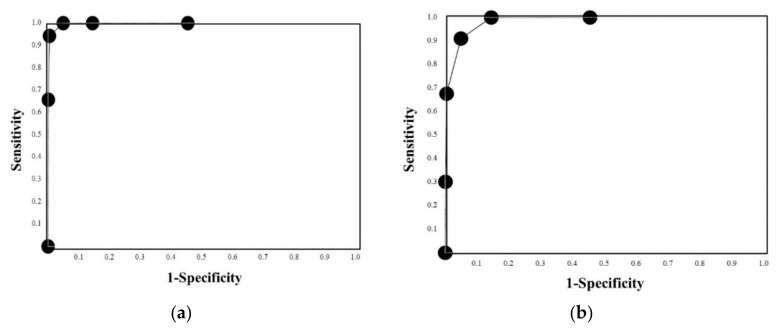
ROC curve of the quick DIC score for DIC vs. non-DIC (**a**), DIC and pre-DIC vs. non-DIC (**b**), or survivor vs. non-survivor (**c**). ROC, receiver operating characteristic; DIC, disseminated intravascular coagulation; area under the curve, (**a**) 0.997, (**b**) 0.984, and (**c**) 0.685.

**Table 1 jcm-11-01028-t001:** Subjects.

Underlying Diseases	Non-DIC	Pre-DIC	DIC	Number of DIC (%)
Sepsis	26	21	11	11/58 (19.0%)
Critical illness without sepsis	95	23	34	34/152 (22.4%)
Pneumonia	78	17	12	12/107 (11.2%)
Obstetric diseases	13	1	2	2/16 (12.5%)
Solid cancer	33	6	3	3/42 (7.1%)
Aneurysm	66	9	3	3/78 (3.8%)
Other respiratory diseases	24	3	1	1/28 (3.6%)
Peripheral arterial and venous thrombosis	39	4	1	1/44 (2.3%)
Hematological disorders	40	0	1	1/41 (2.4%)
Other infections	64	12	1	1/77 (1.3%)
Other digestive disorders	84	2	1	1/87 (1.1%)
Cerebrovascular disorders	174	2	0	0/176 (0%)
Acute coronary syndrome	66	1	0	0/67 (0%)
Other heart diseases	90	4	0	0/94 (0%)
Urinary tract diseases	13	1	0	0/14 (0%)
Unidentified clinical syndrome	125	0	0	0/125 (0%)
Others	51	0	0	0/50 (0%)
Total	1114	109	70	70/1293 (5.4%)

DIC, disseminated intravascular coagulation; Non-DIC, DIC score ≤ 4; Pre-DIC, DIC score 5 or 6; DIC, DIC score ≥ 7 using the Japanese Ministry of Health, Labor and Welfare diagnostic criteria.

**Table 2 jcm-11-01028-t002:** Results of a multiple regression analysis of PT-INR, D-dimer, and platelet counts for diagnosing DIC or pre-DIC vs. non-DIC.

	DIC/Pre-DIC	β	SE (β)	Stdβ	t	N	*p*
PT-INR	DIC	0.09643	0.01059	0.2564	9.10465	1178	<0.000001
	DIC + Pre-DIC	0.09596	0.01589	0.1660	6.04085	1287	<0.000001
PLT	DIC	−0.0034	0.00046	−0.2098	7.36469	1178	<0.000001
	DIC + Pre-DIC	−0.0061	0.00068	−0.2420	8.9452	1287	<0.000001
D-dimer	DIC	0.00379	0.00027	0.3776	13.9969	1178	<0.000001
	DIC + Pre-DIC	0.00501	0.00039	0.3365	12.8174	1287	<0.000001
Sex	DIC	0.02700	0.01144	0.0686	2.35955	1178	0.01846
	DIC + Pre-DIC	0.03496	0.01686	0.0577	2.07386	1287	0.03829
Age	DIC	−0.0007	0.00029	−0.0740	2.54647	1178	0.01101
	DIC + Pre-DIC	−0.0005	0.00043	−0.0359	1.28869	1287	0.19774

The double correlation coefficient, DIC, R = 0.57684 (*p* < 0.000001); Pre-DIC + DIC, R = 0.51303 (*p* < 0.000001); DIC, disseminated intravascular coagulation; Non-DIC, DIC score ≤ 4; Pre-DIC, DIC score 5 or 6; DIC, DIC score ≥ 7 using the Japanese Ministry of Health, Labor and Welfare diagnostic criteria. PT-INR, prothrombin time–international normalized ratio.

**Table 3 jcm-11-01028-t003:** ROC analysis of PT-INR, platelet counts, and D-dimer for diagnosing DIC or pre-DIC vs. non-DIC.

	DIC/Pre-DIC	Cut-Off Value	Sensitivity	Specificity	AUC	NPV	Odds’ Ratio
PT-INR	DIC	1.20	88.8%	88.0%	0.934	98.9%	45.3
	DIC + Pre-DIC	1.11	77.9%	77.9%	0.848	95.7%	12.5
PLT (×10^10^/L)	DIC	12.0	93.6%	80.0%	0.925	44.1%	58.8
	DIC + Pre-DIC	16.4	76.2%	76.2%	0.828	34.0%	10.2
D-dimer (μg/mL)	DIC	10.0	93.1%	92.5%	0.971	99.6%	166.4
	DIC + Pre-DIC	7.8	86.4%	87.0%	0.930	97.9%	42.5

DIC, disseminated intravascular coagulation; Non-DIC, DIC score ≤ 4; Pre-DIC, DIC score 5 or 6; DIC, DIC score ≥ 7 using the Japanese Ministry of Health, Labor and Welfare diagnostic criteria. PT-INR, prothrombin time–international normalized ratio; AUC, area under the curve; NPV, negative predictive value.

**Table 4 jcm-11-01028-t004:** Quick DIC diagnostic criteria.

	Cut-Off Value	Points
PT-INR	≥1.2	1
Platelet count	≤12.0 × 10^10^/L	1
D-dimer	≥10.0 μg/mL	2
Underlying diseases * due to DIC		1
Total	≥3 points	Possible DIC

DIC, disseminated intravascular coagulation; PT-INR, prothrombin time–international normalized ratio; * severe infections, hematological malignancy, solid cancer, aneurysm, obstetric diseases, critical illness such as trauma, shock and inflammation, multiple organ failure, etc.

**Table 5 jcm-11-01028-t005:** Results of the ROC analysis of the quick DIC score for diagnosing DIC or pre-DIC vs. non-DIC.

	Score	Sensitivity	Specificity	PPV	NPV	Likelihood Ratio
DIC AUC, 0.997 Cut-off value, 3	5	65.7%	99.9%	97.9%	94.1%	732.1
	4	94.3%	99.6%	93.0%	99.6%	210.1
	3	100%	95.1%	56.0%	100%	20.3
	2	100%	85.5%	30.3%	100%	6.9
	1	100%	54.6%	12.2%	100%	2.2
	0	100%	0%	-	-	-
DIC + pre-DIC AUC, 0.984 Cut-off value, 2	5	30.2%	99.9%	98.2%	89.9%	480.8
	4	67.6%	99.5%	96.0%	95.0%	462.7
	3	91.1%	95.1%	74.8%	98.5%	18.4
	2	100%	85.5%	52.6%	100%	6.9
	1	100%	54.6%	26.1%	100%	2.2
	0	100%	0%	-	-	-

DIC, disseminated intravascular coagulation; PPV, positive predictive value; NPV, negative predictive value.

## Data Availability

The data presented in this study are available on request from the corresponding author. The data are not publicly available due to privacy restrictions.

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
