# Peer review of "Proposal of Quick Diagnostic Criteria for Disseminated Intravascular Coagulation"

_jcm, 2022, doi:10.3390/jcm11041028_

Round 1

Reviewer 1 Report

In the manuscript the authors propose novel score system for diagnosis and prognosis of disseminated intravascular coagulation.

The manuscript is in overall clearly written and relevant to the field. It is not surprising that quick DIC score  has good accuracy when compared to JMHLW diagnostic criteria. In both tests almost the same parameters are used (PT and platelets), and D-dimers are correlated with FDP. What is important quick DIC score is simplified. 

My major question is why you did you use JMHLW diagnostic criteria not  ISTH, JAAM or JSTH?

Minor remarks are as follows:

  1. I suggest to use disseminated intravascular coagulation in the title.
  2. DIC score on figures is not clear. Why are there four categories? It should be non-DIC, pre-DIC and DIC; ≤4; 5-6; and ≥7 points.
  3. There are typos in references.

Author Response

In the manuscript the authors propose novel score system for diagnosis and prognosis of disseminated intravascular coagulation.

Comment 1.                                                                                                         The manuscript is in overall clearly written and relevant to the field. It is not surprising that quick DIC score has good accuracy when compared to JMHLW diagnostic criteria. In both tests almost the same parameters are used (PT and platelets), and D-dimers are correlated with FDP. What is important quick DIC score is simplified.

Response 1. The most important point concerning the quick DIC system is its use of a point of underlying due to the onset of DIC. The second point is the single cut-off value for PT and D-dimer, which has a high NPV for diagnosing DIC. The third point is the single cut-off value for the platelet counts, which has a high sensitivity for diagnosing DIC. These points have now been added to the Discussion section.

Comment 2.                                                                                                          My major question is why you did you use JMHLW diagnostic criteria not ISTH, JAAM or JSTH?

Response 2. The reason has been now described in the Introduction section.

Minor remarks are as follows:

Comment 3.                                                                                                           1. I suggest to use disseminated intravascular coagulation in the title.

Response 3. The title has been now revised.

Comment 4.                                              2. DIC score on figures is not clear. Why are there four categories? It should be non-DIC, pre-DIC and DIC; ≤4; 5-6; and ≥7 points.

Response 4. The figures have been revised.

Comment 5.                                                                                                      There are typos in references.

Response 5. The references have been revised.

Reviewer 2 Report

The idea of ​​generating a rapid scoring system as a tool for the rapid diagnosis of DIC is interesting and can be really useful in clinical practice. The selection of the patient groups and the parameters evaluated in the study are adequate. However, the methodology presents unclear and/or improvable aspects that must be resolved before considering that the article is ready for publication.

Methods: In addition to describing the values ​​and interquartile ranges of the variables in the groups of patients, the reference values ​​in the general population (healthy people) should also be indicated. It is not explained how the cut-off point is calculated from the ROC curves: Was it using the Youden index? Was it prioritizing sensitivity? was prioritizing specificity? This information should be incorporated in the methods section.

Indicate if any statistical treatment software was used and cite it

Results: Before proposing the predictive model, it is necessary to know the real strength of the qualitative variables DD-elevated, thrombocytopenia and high INR (patients who are above the cut-off points defined in the study). The Odds ratio (with 95% CI) for the different outputs (DIC and Pre-DIC) must be indicated.

It is not clearly explained why a score of 1 is assigned to the positivity of each variable. It would be expected that the participation of each one of the variables in the complete model would be different. For example, one would expect that high levels of DD would be a more powerful risk factor than the rest for the appearance of DIC (example, 3 points), that thrombocytopenia would have an intermediate value (2 points) and that an elevated INR would have a lower impact (1 point), however a score of one is applied to all of them.

It is necessary to establish a model with scores closer to the real strength of the risk factor. This model would be just as easy to handle in the clinic and would provide more robust information. To assess the real participation of each variable in a predictive risk model, a multivariable logistic regression analysis should be performed that includes all the variables related to DIC risk. From this analysis, a more accurate and robust predictive model can be made. For this task, authors are advised to consult with statistical experts.

I have not been able to access the supplementary material

Author Response

The idea of ​​generating a rapid scoring system as a tool for the rapid diagnosis of DIC is interesting and can be really useful in clinical practice. The selection of the patient groups and the parameters evaluated in the study are adequate.               However, the methodology presents unclear and/or improvable aspects that must be resolved before considering that the article is ready for publication.

Response 1 The methods section has been revised.

Comment 2.                                                                                               Methods: In addition to describing the values ​​and interquartile ranges of the variables in the groups of patients, the reference values ​​in the general population (healthy people) should also be indicated. It is not explained how the cut-off point is calculated from the ROC curves: Was it using the Youden index? Was it prioritizing sensitivity? was prioritizing specificity? This information should be incorporated in the methods section. Indicate if any statistical treatment software was used and cite it

Response 2. The cut-off value was determined as the point at which the sensitivity curve and the specificity curve intersected. The methods section has been revised. The software program has been shown in the Statistical analyses section.                                                    

Comment 3.                                                                                                           Results: Before proposing the predictive model, it is necessary to know the real strength of the qualitative variables DD-elevated, thrombocytopenia and high INR (patients who are above the cut-off points defined in the study). The Odds ratio (with 95% CI) for the different outputs (DIC and Pre-DIC) must be indicated.

Response 3. The odds ratio has been added.

Comment 4.                                                                                                            It is not clearly explained why a score of 1 is assigned to the positivity of each variable. It would be expected that the participation of each one of the variables in the complete model would be different. For example, one would expect that high levels of DD would be a more powerful risk factor than the rest for the appearance of DIC (example, 3 points), that thrombocytopenia would have an intermediate value (2 points) and that an elevated INR would have a lower impact (1 point), however a score of one is applied to all of them.

Response 4. After a reevaluation of the usefulness of biomarkers for diagnosing DIC or pre-DIC, we changed one point to two points for the D-dimer score.

Comment 5. It is necessary to establish a model with scores closer to the real strength of the risk factor. This model would be just as easy to handle in the clinic and would provide more robust information. To assess the real participation of each variable in a predictive risk model, a multivariable logistic regression analysis should be performed that includes all the variables related to DIC risk. From this analysis, a more accurate and robust predictive model can be made. For this task, authors are advised to consult with statistical experts.

Response 5. A multivariable logistic regression analysis is now shown in Table 2.

Comment 6.                                                                                                             I have not been able to access the supplementary material

Response 6. The supplementary table has been uploaded.

Round 2

Reviewer 2 Report

ok